# MoDiff: a Protein Diffusion Model for Multi-motif Scaffolding Problem

## Abstract

The motif scaffolding problem, crucial in drug and enzyme design, involves creating a scaffold to support one or more specified motifs. Existing methods employ the paradigm of inpainting, thereby limiting their applicability to scenarios where only a single motif is present or the positions between multiple motifs are known. However, in many significant scenarios, the positions between motifs are unknown, leaving the multi-motif scaffolding problem open. To tackle this challenge, we introduce a protein diffusion model called MoDiff. During the diffusion process, MoDiff implicitly assigns motifs to the protein backbone, thereby achieving the automatic design of relative positions among motifs. Our experiments demonstrate that MoDiff can: 1) solve the multi-motif scaffolding problem even when the positions between motifs are unknown, and 2) generate diverse scaffolds based on multiple given motifs. This indicates that MoDiff is a potential general solution to the multi-motif scaffolding problem.

## 1 Introduction

Protein design plays a crucial role in drug development and enzyme design, often abstracted into the motif-scaffolding problem (Procko et al., 2014; Correia et al., 2014; Jiang et al., 2008; Siegel et al., 2010). Here, 'motif' refers to specific functional segments of protein structures. The objective of this problem is to devise a scaffold that supports designated motifs, thereby imparting the desired functionality to the designed protein. Recently, protein design approaches based on diffusion models (Song et al., 2021) have made significant strides in generating scaffolds for single motif. For instance, RFdiffusion (Watson et al., 2023), employing the inpainting paradigm, treats the given motif as the known part of the protein, with the remaining amino acids treated as inpainting content to generate the corresponding scaffold.

However, existing methods face significant limitations when applying the inpainting paradigm to the design of scaffolds for multiple motifs (introduced later). At the same time, the multi-motif scaffolding problem holds paramount importance in the design of antibodies and enzymes (Sec.2.1), as proteins frequently contain more than one motif. Consequently, identifying a general solution for the multi-motif scaffolding problem holds great significance.

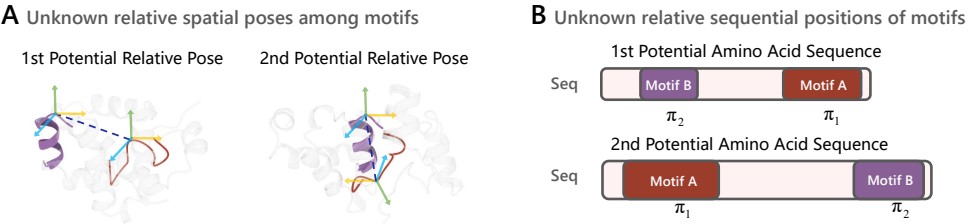

Figure 1: **Challenges of multi-motif scaffolding problem. A)** illustrating the uncertainty in the spatial relationship between motif A and motif B , presenting various possible relative poses. **B)** depicting the undefined amino acid correspondence $\pi_1, \pi_2$ between the motif and the protein.

The limitations of existing methods in addressing the multi-motif scaffolding problem can be primarily attributed to two aspects, as shown in Fig. 1. On one hand, the inpainting paradigm (Watson et al., 2023) *necessitates a defined and rational relative positioning* of motifs, enabling the generation of a scaffold by treating multiple motifs as a "single entire motif". On the other hand, in many multi-motif scaffolding problems (Sec. 2.1), *the positions between motifs are unknown* and need to be designed, posing a challenge for existing methods. This challenge persists even in the presence of alternative motif-scaffolding approaches like Chroma (Ingraham et al., 2023) and SMCDiff (Trippe et al., 2022). As a result, addressing the multi-motif scaffolding problem still demands extensive experimentation and expert knowledge.

In this paper, we introduce a motif-conditioned generation model called MoDiff. MoDiff automatically establishes the sequential and spatial relative positions between motifs, addressing the challenge of unknown motif positions and providing the capability to generate diverse and designable scaffolds. The core concept of MoDiff differs from inpainting. Through learning from data distributions, the MoDiff network guides motifs to rational positions on the protein backbone during the diffusion process, achieving the automatic design of relative positions among motifs.

However, we observed that traditional cross-attention methods are ineffective in guiding motif generation. To address this issue, we propose an implicit matching module. Specifically, during training, as we utilize randomly cropped segments from the protein as virtual motifs, the correspondence between motifs and the protein backbone is known. Consequently, within this module, this correspondence is leveraged to provide matching supervision between motifs and the protein backbone, guiding the model in placing motifs at reasonable positions along the backbone. Additionally, we introduce a motif reconstruction loss to further guide motif generation, ultimately enhancing the success rate of scaffold design.

*In silico* experiments demonstrate that MoDiff, when confronted with diverse motifs, exhibits the capability to 1) effectively design the positions between motifs and 2) generate diverse and designable scaffolds. Interestingly, we observe a significant increase in the novelty of proteins generated by MoDiff compared to the unconditioned baseline, VFN-Diff (Mao et al., 2023). This phenomenon arises because, in cases where specified motifs are sourced from different proteins, there may not be any known protein containing the given motifs simultaneously. Consequently, the proteins designed by MoDiff with a higher probability differ from all known proteins. Next, ablation experiments indicate the effectiveness of the motif reconstruction loss and implicit matching module, significantly enhancing the success rate of design. Those suggest that MoDiff represents a potential general solution to the multi-motif scaffolding problem. The contributions of this study can be summarized as follows:

- We highlight the limitations of existing methods in addressing the multi-motif scaffolding problem and showcase the promising prospects of deep learning approaches in this domain.
- We introduce MoDiff, a method capable of efficiently designing proteins with multiple motifs. Experimental results support MoDiff as a potential general solution to the multi-motif scaffolding problem.
- The proposed implicit matching module and motif reconstruction loss effectively enhance the success rate of design.

## 2 RELATED WORK

### 2.1 MULTI-MOTIF SCAFFOLDING PROBLEM

Many protein design problems are formulated as the multi-motif scaffolding problem (Cao et al., 2022; Wang et al., 2022; Yang et al., 2021; Sesterhenn et al., 2020; Linsky et al., 2020; Wang et al., 2022). For instance, 1) a protein (Roy et al., 2023) with high selectivity is designed by incorporating multiple known binding motifs, and 2) through expert knowledge, two EF-hand motifs are successfully integrated into a protein (Roel-Touris et al., 2023) to enhance enzyme activity. Notably, in many cases (Roy et al., 2023; Roel-Touris et al., 2023; Davila-Hernandez et al., 2023; Jiang et al., 2023), at least one position, either in terms of spatial or sequential relative positions between motifs, remains uncertain. While some problems under this circumstance can be addressed, it still requires extensive trial and error, human effort, and expert knowledge. Moreover, these methods exhibit significant

limitations (Gelman & Gruebele, 2014; Tischer et al., 2020; Wu et al., 2021; Tischer et al., 2020; Huang et al., 2022a), especially when faced with situations lacking suitable templates and references in the Protein Data Bank (PDB). In contrast, MoDiff presents a potential general solution that does not necessitate expert knowledge.

## 2.2 DIFFUSION MODELS FOR MOTIF SCAFFOLDING

With the emergence of numerous protein diffusion models (Wu et al., 2022; Yim et al., 2023; De Bortoli et al., 2022; Huang et al., 2022b; Anand & Achim, 2022; Lee et al., 2023; Lisanza et al., 2023; Martinkus et al., 2023), protein design has witnessed a revolutionary paradigm shift. Motif-scaffolding stands as a central task in protein design, with various diffusion model methods addressing this challenge over time. In SMCDiff (Trippe et al., 2022), the authors initially train an unconditionally generated protein backbone diffusion model, ProtDiff. Subsequently, they introduce guidance for motif generation based on ProtDiff. Chroma (Ingraham et al., 2023) adopts a similar paradigm, further incorporating various types of guidance. RFdiffusion (Watson et al., 2023), on the other hand, employs a different inpainting paradigm, treating motifs as given content and generating scaffolds as inpainting components. However, these methods face a common limitation when designing scaffolds to support multiple motifs: the inability to design the positions between motifs, requiring predetermined rational positions for motifs. Nevertheless, as mentioned, determining the positions between motifs is often challenging, restricting the applicability of these methods.

## 3 PRELIMINARIES

### 3.1 PARAMETERIZED REPRESENTATION

Proteins typically consist of $N$ amino acids, and the backbone of each amino acid is composed of four atoms: $C, C_\alpha, O, N$. Following the approach of AlphaFold 2 (Jumper et al., 2021), a deterministic frame $T \in SE(3)$ can be established for the atoms $C, C_\alpha$ and $N$ through the Gram–Schmidt process. Consequently, $i$-th amino acid can be represented by a backbone frame $T_i \in SE(3)$, and the structure of the protein backbone is parameterized as N frames $\mathbf{T} = [T_1, T_2, ..., T_N] \in SE(3)^N$. In previous protein studies, specific segments with distinct biological functions were identified. Any of these segments can be referred to as a motif, denoted as $\mathbf{T}^\mathcal{M} = [T_1^\mathcal{M}, T_2^\mathcal{M}, ..., T_M^\mathcal{M}] \in SE(3)^M$. Here, $M$ represents the number of amino acids in a motif. As the motif is a segment within the protein $\mathbf{T}$, an alignment permutation $\pi$ of the protein $\mathbf{T}$ exists such that $\mathbf{T}^\mathcal{M} = \mathbf{T}_\pi$. Starting from this point, $\pi$ represents a specific index permutation of amino acids.

### 3.2 MULTI-MOTIF SCAFFOLDING PROBLEM

Given the structures of multiple motifs, the objective of multi-motif scaffolding is to sample a protein of length $N$ that incorporates the given motifs. For the sake of brevity, we will describe the case of having two motifs from this point onward, but these concepts can be straightforwardly extended to scenarios with more motifs.

**Definition 1.** (Multi-motif Scaffolding Problem). Given two motifs $\mathbf{T}^{\mathcal{M}_1} \in SE(3)^{M_1}, \mathbf{T}^{\mathcal{M}_2} \in SE(3)^{M_2}$, the task is to sample a protein structure $\hat{\mathbf{T}} = [\hat{T}_1, \hat{T}_2, ..., \hat{T}_N] \in SE(3)^N$ from the conditional probability distribution $p_\theta(\hat{\mathbf{T}}|\mathbf{T}^{\mathcal{M}_1}, \mathbf{T}^{\mathcal{M}_2})$, where $\hat{\mathbf{T}}$ contains the motifs within the protein structures. Here, $M_1$ and $M_2$ represent the numbers of amino acids in the two motifs, respectively.

**Intractability.** The structure of proteins must satisfy the natural constraints, meaning not all protein structures are physically viable. This necessitates the proper arrangement between motifs $\mathbf{T}^{\mathcal{M}_1}$ and $\mathbf{T}^{\mathcal{M}_2}$; otherwise, the corresponding protein $\hat{\mathbf{T}}$ may not exist. However, 1) $\mathbf{T}^{\mathcal{M}_1}$ and $\mathbf{T}^{\mathcal{M}_2}$ are respectively defined with respect to two distinct global frames $T^{g1}, T^{g2} \in SE(3)$, meaning that the reasonable spatial relationship between $\mathbf{T}^{\mathcal{M}_1}$ and $\mathbf{T}^{\mathcal{M}_2}$ is unknown and must be systematically designed. 2) The amino acids correspondence $\pi$ between the motif and the protein is also unknown, requiring a method to establish this correspondence. For instance, the design of a rational slice index $\pi_1$ is necessary to ensure that motif $\mathbf{T}^{\mathcal{M}_1}$ aligns appropriately with a segment of protein $\hat{\mathbf{T}}$, simultaneously ensuring the existence of the protein. Similarly, a corresponding $\pi_2$ also needs to be designed for motif $\mathbf{T}^{\mathcal{M}_2}$.

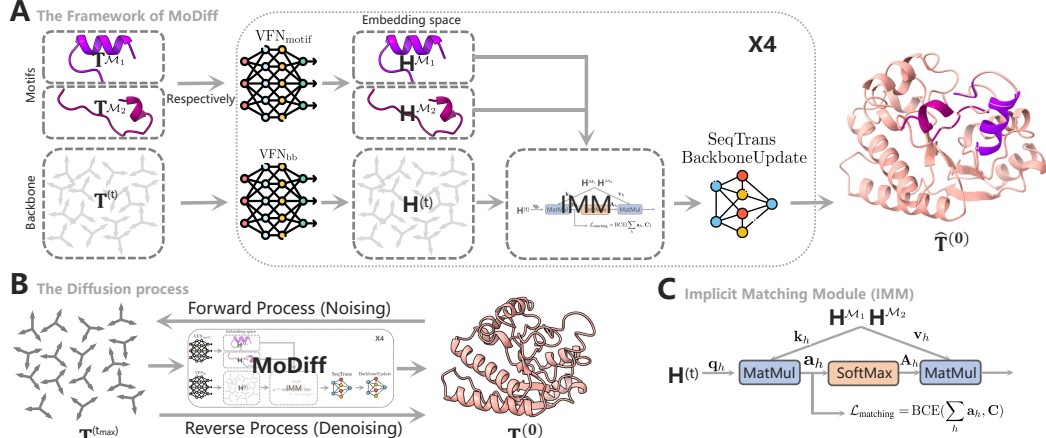

Figure 2: **The MoDiff pipeline. A)** The framework of MoDiff. $\mathbf{T}^{\mathcal{M}_1}$ and $\mathbf{T}^{\mathcal{M}_2}$ represent the given motifs. $\mathbf{T}^{(t)}$ denotes the protein backbone with noise, depicted using frames (Sec.4.2). **B)** The diffusion process, with $t_{\max}$ indicating the maximum number of noise-injected steps. $\mathbf{T}^{(t_{\max})}$ represents pure noise (Sec.3.3). **C)** Implicit Matching Module, responsible for matching the provided motifs to the given backbone(Sec.4.3).

**Evaluation.** To assess the presence of a specified motif, $e.g.\mathbf{T}^{\mathcal{M}_1}$, in the generated protein, we utilize the root mean squared distances (RMSD) of the motif, denoted as $\mathrm{RMSD}_{\mathrm{motif}}$, to measure the structural similarity between the generated structure and the provided motif structure. If $\mathrm{RMSD}_{\mathrm{motif}}$ is $< 1$ Å, we consider the generated structure to contain the specified motif, as 1Å represents atom-level resolution.

**Definition 2.** (Motif RMSD). $\mathrm{RMSD}_{\mathrm{motif}}$ represents the RMSD between a motif, $e.g.\mathbf{T}^{\mathcal{M}_1}$, and a generated protein, $e.g.\hat{\mathbf{T}}$, after finding an optimal matching alignment (Ingraham et al., 2023). This optimal alignment encompasses both spatial and sequential alignments. Spatial alignment involves identifying the optimal transformation $T^{\mathrm{a}}$ aligning the motif with the protein. Sequential alignment entails finding the optimal aligning indices $\pi^{\mathrm{a}}$ on residue sequences among all possible permutations $\Pi$. This can be formulated as:

$$\mathrm{RMSD}_{\mathrm{motif}} = \min_{\pi^{\mathrm{a}} \in \Pi} \min_{T^{\mathrm{a}} \in \mathrm{SE}(3)} \frac{\|\mathbf{x}_{\mathcal{M}} - T^{\mathrm{a}} \circ \hat{\mathbf{x}}_{\pi^{\mathrm{a}}}\|}{\sqrt{M}} \tag{1}$$

where $\hat{\mathbf{x}} \in \mathbb{R}^{N \times 3}$ represents the coordinates of all $C_\alpha$ atoms in the generated protein $\hat{\mathbf{T}}$, and the coordinates of the $C_\alpha$ atom in the $i$-th amino acid are the origin of the amino acid frame $\hat{T}_i$, $i.e.$the translation component of the frame. Similarly, $\mathbf{x}_{\mathcal{M}} \in \mathbb{R}^{M \times 3}$ is the coordinates of all $C_\alpha$ atoms in the motif, $i.e.$the origin of $\mathbf{T}^{\mathcal{M}}$. $\pi^{\mathrm{a}}$ is a set of indices that extracts the elements corresponding to the motif $\mathbf{x}_{\mathcal{M}}$ from the protein $\hat{\mathbf{x}}$. $\circ$ represents a transformation operation.

### 3.3 SE(3) DIFFUSION MODEL FOR PROTEIN GENERATION

In our work, we adopt the paradigm of SE(3) diffusion (Yim et al., 2023), and we briefly review it here. For more detailed information, please refer to the original paper. The goal of SE(3) diffusion is to model the distribution of the data on the SE(3) manifold. This method is built upon the Riemannian (Do Carmo & Flaherty Francis, 1992) score-based generative modeling approach (De Bortoli et al., 2022; Huang et al., 2022b), where a score network is trained to minimize a denoising score matching (DSM) loss:

$$\mathcal{L}(\theta) = \mathbb{E}_{t \in \mathcal{U}(0,T^{\mathrm{s}})} \mathbb{E}_{p_t(\mathbf{X})} [\lambda_t \| \nabla_{\mathbf{X}} \log p_t(\mathbf{X}) - \mathbf{s}_\theta(\mathbf{X},t) \|_2^2] \tag{2}$$

Here, $t$ represents the diffusion time step, and $T^{\mathrm{s}}$ is the maximum time step. $\lambda_t$ denotes a weight associated with time $t$. In this context, the uppercase $\mathbf{X}$ represents a random variable, distinct from the lowercase $\mathbf{x}$ used earlier to denote the coordinates of $C_\alpha$ atoms. The score network $\mathbf{s}_\theta$ represents a neural network encoding the protein structure. In this work, we start by utilizing VFN (Mao et al.,

2023) as the neural network encoder $\mathbf{s}_\theta$. The input to this encoder is the structure $\mathbf{T}^{(t)}$ with the $t$-th step noise, and its output is the denoised structure $\hat{\mathbf{T}}^{(0)}$:

$$(\hat{\mathbf{T}}^{(0)}, \hat{\boldsymbol{\psi}}) = \mathrm{VFN}(\mathbf{T}^{(t)}, t) \tag{3}$$

where $\hat{\boldsymbol{\psi}}$ is a group of parameters used to compute the angle for the position of atomic O.

## 4 METHODS

As shown in Fig. 2, MoDiff addresses the multi-motif scaffolding problem by introducing the motif conditions as inputs to the neural network $\mathbf{s}_\theta$. This can be formulated as:

$$(\hat{\mathbf{T}}^{(0)}, \hat{\boldsymbol{\psi}}) = \mathrm{MoDiff}(\mathbf{T}^{(t)}, \mathbf{T}^{\mathcal{M}_1}, \mathbf{T}^{\mathcal{M}_2}, t) \tag{4}$$

The motivation is straightforward: For previous methods, *e.g.*SMCDiff (Trippe et al., 2022), it is necessary to establish the positional relationships between motifs reasonably before providing the motifs guidance; however, the positions between motifs are sometimes uncertain, requiring a significant amount of expert knowledge. In contrast, in this way, MoDiff can guide the rational arrangement of motifs by *leveraging prior knowledge learned from the data distribution*, eliminating the need for expert knowledge.

In the subsequent sections, we first present the methodology for constructing a training dataset (Sec. 4.1) with multiple motifs. Subsequently, we outline the overall framework (Sec. 4.2) of MoDiff, consisting of two crucial components: the Implicit Matching Module (Sec. 4.3) and the Motif Reconstruction Loss (Sec. 4.4). Finally, we provide a concise overview of the inference process (Sec. **??**) employed by MoDiff.

### 4.1 MULTI-MOTIF TRAINING SET

Due to the fact that motifs can represent any arbitrary segment of a protein, we propose a random crop method to construct a virtual multi-motif training set. Specifically, given a known protein structure $\mathbf{T} \in \mathrm{SE}(3)^N$, a $\mathrm{RandomCrop}$ function randomly extracts two segments from the protein $\mathbf{T}$ and returns the corresponding indices $\pi_1 \in \{1, 2, ...N\}^{M_1}, \pi_2 \in \{1, 2, ...N\}^{M_2}$. With $\pi_1, \pi_2$, the virtual motifs, $\mathbf{T}^{\mathcal{M}_1} \in \mathrm{SE}(3)^{M_1}, \mathbf{T}^{\mathcal{M}_2} \in \mathrm{SE}(3)^{M_2}$, on the protein $\mathbf{T}$ can be obtained through indexing operations. This can be expressed as:

$$\begin{aligned} (\pi_1, \pi_2) &= \mathrm{RandomCrop}(\mathbf{T}) \\ (\mathbf{T}^{\mathcal{M}_1}, \mathbf{T}^{\mathcal{M}_2}) &= (\mathbf{T}_{\pi_1}, \mathbf{T}_{\pi_2}) \end{aligned} \tag{5}$$

### 4.2 FRAMEWORK OF MODIFF

MoDiff consists of two branches: the motif branch and the protein backbone branch. These branches are employed to encode the geometric information of motifs, $\mathbf{T}^{\mathcal{M}_1}, \mathbf{T}^{\mathcal{M}_2}$, and the protein backbone $\mathbf{T}^{(t)}$, respectively. Furthermore, within the protein backbone branch, the representation of motifs is aggregated into the protein backbone representation and utilized to predict the denoised structure $\hat{\mathbf{T}}^{(0)}$.

**The motif branch.** The two motifs, $\mathbf{T}^{\mathcal{M}_1}, \mathbf{T}^{\mathcal{M}_2}$, are separately fed into a same geometric encoder to obtain corresponding $d$-dimensional geometric features $\mathbf{H}^{\mathcal{M}_1} \in \mathbb{R}^{M_1 \times d}$ and $\mathbf{H}^{\mathcal{M}_2} \in \mathbb{R}^{M_2 \times d}$. Here, we employ a standard VFN network, $\mathrm{VFN}_{\mathrm{motif}}$, as the geometric encoder, treating each amino acid as a node for encoding. This can be written as:

$$\mathbf{H}^{\mathcal{M}_i} = \mathrm{VFN}_{\mathrm{motif}}(\mathbf{T}^{\mathcal{M}_i}) \tag{6}$$

**The protein backbone branch.** During training, the protein $\mathbf{T}$ is randomly noised at time step $t$ and then used as the input $\mathbf{T}^{(t)}$ for the protein backbone branch. This backbone $\mathbf{T}^{(t)}$ is similarly encoded using a standard VFN network $\mathrm{VFN}_{\mathrm{bb}}$ to obtain corresponding geometric features $\mathbf{H} \in \mathbb{R}^{N \times d}$, written as:

$$\mathbf{H} = \mathrm{VFN}_{\mathrm{bb}}(\mathbf{T}^{(t)}) \tag{7}$$

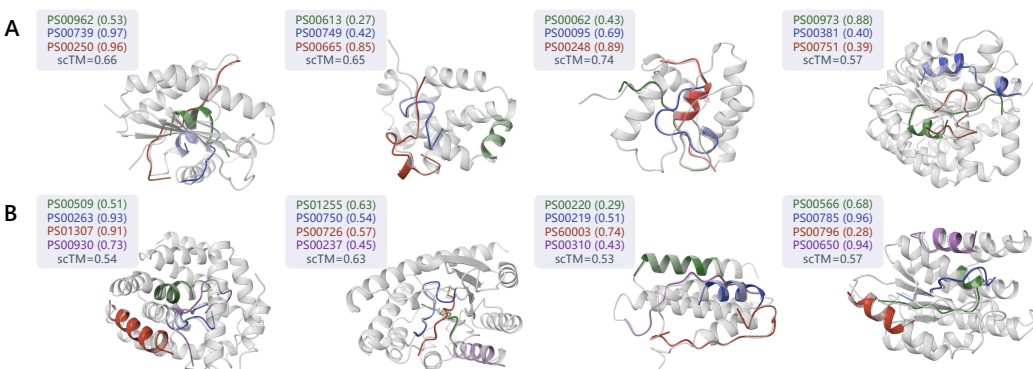

Figure 3: **Visual results of 3 and 4 motifs on $\mathcal{M}^{\mathbf{unknown}}$.** Different motifs are distinguished by different colors, with the color of each motif corresponding to the color of its respective motif ID and $\mathrm{RMSD}_{\mathrm{motif}}$. The motif ID and $\mathrm{RMSD}_{\mathrm{motif}}$ are displayed in the format 'motif ID ($\mathrm{RMSD}_{\mathrm{motif}}$)' at the top right corner of each sample. Here, scTM represents the protein's designability, and $\mathrm{RMSD}_{\mathrm{motif}}$ represents the error between the given motif and the generated structure's motif. A design is considered successful if $\mathrm{scTM} > 0.5$ and $\mathrm{RMSD}_{\mathrm{motif}} < 1$. A) Results with 3 given motifs. B) Results with 4 given motifs.

Next, a component called the Implicit Matching Module (introduced later), IMM, aggregates the representations $\mathbf{H}^{\mathcal{M}_1}$ and $\mathbf{H}^{\mathcal{M}_2}$ of motifs into $\mathbf{H}$:

$$\mathbf{H} \leftarrow \mathrm{IMM}(\mathbf{H}, \mathbf{H}^{\mathcal{M}_1}, \mathbf{H}^{\mathcal{M}_2}) \tag{8}$$

The updated representation $\mathbf{H}$ is then encoded by a sequence transformer, SeqTrans, and subsequently fed into a backbone update module, BackboneUpdate, to predict the denoised structure $\hat{\mathbf{T}}^{(0)}$ and the angle $\hat{\psi}$:

$$\mathbf{H} \leftarrow \mathrm{SeqTrans}(\mathbf{H})$$
$$(\hat{\mathbf{T}}^{(0)}, \hat{\psi}) = \mathrm{BackboneUpdate}(\mathbf{H}, \mathbf{T}^{(t)}) \tag{9}$$

Here, SeqTrans and BackboneUpdate are standard modules in FrameDiff. The protein backbone branch will be iteratively repeated four times, with the output of each round serving as the input for the next iteration. Between each iteration, the model does not share parameters.

### 4.3 IMPLICIT MATCHING MODULE

The goal of the Implicit Matching Module is to aggregate the motif features of $\mathbf{H}^{\mathcal{M}_1}, \mathbf{H}^{\mathcal{M}_2}$ into the backbone representation $\mathbf{H}$. Therefore, the most straightforward approach is to use cross attention (Vaswani et al., 2017). However, we found that traditional cross attention is not effective in aggregating these features because it overlooks the correspondence $(\pi_1, \pi_2)$ between motifs and the protein backbone. Thus, we introduce explicit $\pi$-related supervision on top of cross attention. This allows the attention module to implicitly match motifs and amino acids of the protein when aggregating features.

We begin by briefly revisiting the standard cross-attention mechanism. Taking the cross attention between motif $\mathbf{H}^{\mathcal{M}_1}$ and protein $\mathbf{H}$ as an example, the $h$-th head in cross attention utilizes three linear layers, denoted as $\phi_{\mathrm{q}}$, $\phi_{\mathrm{k}}$, and $\phi_{\mathrm{v}}$, to transform $\mathbf{H}$ into $\mathbf{q}_h \in \mathbb{R}^{N \times d_k}$ and $\mathbf{H}^{\mathcal{M}_1}$ into $\mathbf{k}_h, \mathbf{v}_h \in \mathbb{R}^{M_1 \times d_k}$. Here, $d_k$ represents the dimensionality of features. Subsequently, a similarity matrix, denoted as $\mathbf{a}_h \in \mathbb{R}^{N \times M_1}$, is obtained through the dot product of $\mathbf{q}_h$ and $\mathbf{k}_h$. Following this, the matrix $\mathbf{a}_h$ undergoes softmax operation to yield the final attention map $\mathbf{A}_h \in \mathbb{R}^{N \times M_1}$. Using $\mathbf{A}_h$, the values $\mathbf{v}_h$ are aggregated. Finally, the outputs from different heads are concatenated to obtain the final output $\mathbf{O} \in \mathbb{R}^{N \times d_k}$. This can be written as:

$$\mathbf{a}_h = \frac{\mathbf{q}_h \mathbf{k}_h^{\top}}{\sqrt{d_k}}$$
$$\mathbf{A}_h = \mathrm{Softmax}(\mathbf{a}_h) \tag{10}$$
$$\mathbf{O} = \mathrm{Linear}(\mathrm{Concat}_h(\mathbf{A}_h \mathbf{v}_h))$$

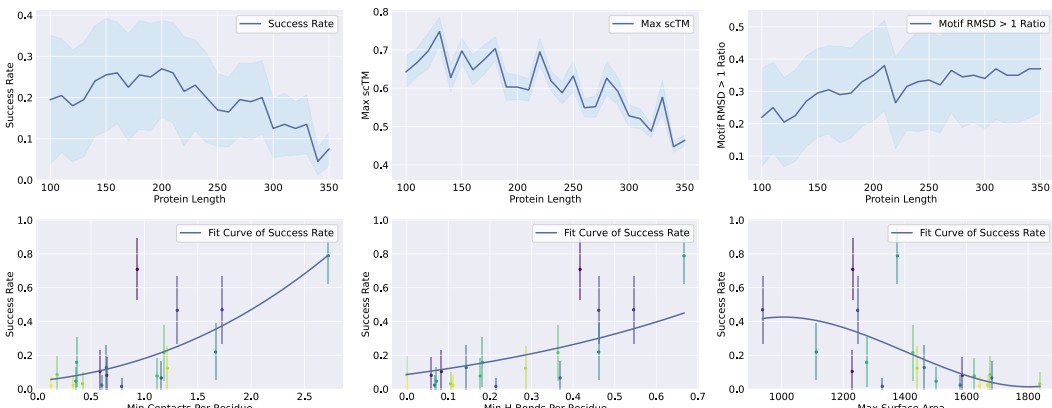

Figure 4: **Deep analysis of the success rate *w.r.t* motif features.** We conduct separate analyses on protein length, hydrogen bonds number per residue, contacts number per residue, and solvent-accessible surface area to assess their impact on the design success rate and various influencing factors.

Here, $\text{Concat}_h$ denotes the concatenation of results from all heads, and $\text{Linear}$ represents a linear layer.

Next, we introduce matching supervision on $\mathbf{a}_h$. Specifically, each element in $\mathbf{a}_h$ can be regarded as the residue-matching score between the motif and the protein backbone. Consequently, we transform $\pi_1$ into a binary one-hot encoding $\mathbf{C} \in \mathbb{R}^{N \times M_1}$ representing whether a pair of amino acids match, through a binary $\text{OneHot}$ operation. Subsequently, $\mathbf{C}$ is used to supervise $\mathbf{a}_h$ via a Binary Cross Entropy loss, $\text{BCE}$, enabling the model to learn the matching relationship between motifs and the protein. Importantly, we refrain from directly supervising $\mathbf{a}_h$ to avoid constraining the model's expressive capability. Instead, we supervise the sum of $\mathbf{a}_h$ across all heads, mitigating the aforementioned issue:

$$\mathbf{C}_i = \text{OneHot}\left(\pi_i\right)$$

$$\mathcal{L}_{\text{matching}} = \sum_i \text{BCE}\left(\sum_h \mathbf{a}_h, \mathbf{C}_i\right) \tag{11}$$

### 4.4 Motif Reconstruction Loss

We adopted the standard SE(3) diffusion loss, Eq.2, for guiding the model during training. However, this loss does not effectively guide the model in generating structures with specified motifs. To address this challenge, we introduce the Motif Reconstruction loss, denoted as $\mathcal{L}_{\text{MR}}$, on top of the standard loss, Eq.2. Specifically, since during training, the correspondence $\pi_1, \pi_2$ between motifs and the protein is known, we use $\pi_1, \pi_2$ to extract the amino acids corresponding to the motifs, resulting in $\hat{\mathbf{T}}_{\pi_1}^{(0)}, \hat{\mathbf{T}}_{\pi_1}^{(0)}$. The optimization objective is to minimize the error between $(\hat{\mathbf{T}}_{\pi_1}^{(0)}, \hat{\mathbf{T}}_{\pi_1}^{(0)})$ and the actual motifs $(\mathbf{T}^{\mathcal{M}_1}, \mathbf{T}^{\mathcal{M}_2})$. This can be formulated as:

$$\mathcal{L}_{\text{MR}} = \text{FAPE}(\mathbf{T}^{\mathcal{M}_1}, \hat{\mathbf{T}}_{\pi_1}^{(0)}) + \text{FAPE}(\mathbf{T}^{\mathcal{M}_2}, \hat{\mathbf{T}}_{\pi_2}^{(0)}) \tag{12}$$

Here, FAPE represents the Frame Aligned Point Error, proposed by AlphaFold 2 (Jumper et al., 2021), which is used to quantify the error between two structures.

## 5 Experiments

In this section, we investigate two key questions through a series of experiments. 1) Whether MoDiff can effectively solve the multi-motif scaffolding problem without the need for expert knowledge and generate diverse scaffolds (Sec.5.3). 2) Whether the proposed enhancement module, IMM and $\mathcal{L}_{\text{MR}}$, is effective (Sec.5.4). Before addressing these questions, we first introduce the training details (Sec.5.1) of our model and the *in silico* evaluation details (Sec.5.2).

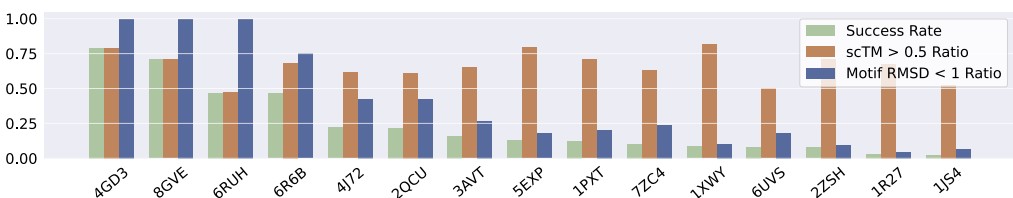

Figure 5: **Metrics across design tasks on $\mathcal{M}^{\text{exist}}$.** The figures illustrate the success rate, the ratio of scTM > 0.5, and the ratio of $\text{RMSD}_{\text{motif}} < 1$ for each task, respectively. The x-axis represents the Protein Data Bank (PDB) ID of the proteins.

## 5.1 TRAINING DETAILS

To construct the training set, we employ the approach used in FrameDiff to select proteins meeting specific criteria from the Protein Data Bank (Berman et al., 2000). The criteria requires protein lengths to fall within the range of 60 to 512 amino acids and have a resolution lower than 5Å. This selection process results in a total of 21,399 proteins. Throughout the training procedure, pairs of virtual motifs, each with lengths ranging from 10 to 20 amino acids, are randomly extracted from the protein backbones (Eq.5) and subsequently input into the model. To speed up the training, we utilize VFN-Diff (Mao et al., 2023) as pretraining weights and conducted training for 250,000 iterations.

## 5.2 EVALUATION

**Motif test samples.** The PROSITE dataset (Sigrist et al., 2012) contains 16,251 authentic motifs, updated as of January, 2024. Therefore, we utilize PROSITE as our motif testing data source but excluded those motifs present in our training set. Based on the filtered data, we randomly select 2 to 4 motifs to evaluate MoDiff. For a more detailed analysis and ablation studies, we choose 2 motifs as the primary experimental setting. In this setting, all double motif test samples are categorized into two classes: $\mathcal{M}^{\text{unknown}}$ and $\mathcal{M}^{\text{exist}}$. A total of 52 motifs, with lengths ranging from 10 to 20 residues, are randomly selected from PROSITE to analyze the impact of different motif combinations on the final performance. $\mathcal{M}^{\text{unknown}}$ represents test samples where the two given motifs come from different proteins. In this case, it is unknown whether there exists a protein that can simultaneously contain the two given motifs. $\mathcal{M}^{\text{exist}}$ represents test samples where the two given motifs both come from a same protein. This implies that there is certainly a protein capable of containing both given motifs simultaneously. For comprehensive benchmark details, please refer to Tab. 6 and Tab. 5 in the appendix.

**Protein sampling methods.** Unless specified otherwise, MoDiff adopts the same inference settings as FrameDiff (Yim et al., 2023) and VFN-Diff (Mao et al., 2023), with a noise scale of 0.1 and 500 diffusion steps. Similarly, due to potential variations in MoDiff's performance across different protein lengths, we conduct experiments for each given motif within the protein length range of 100 to 350, with increments of 10. For each length, 10 protein structures are generated, resulting in a total of 260 proteins for each motifs samples. Subsequently, based on these 260 samples, success rates and other relevant metrics are calculated, as elaborated upon later.

**Metric.** Similar to SMCDiff (Trippe et al., 2022) and SE(3) diffusion, the main evaluation metrics include motif RMSD, scTM, success rate, diversity, and novelty. 'scTM$_{0.5}$' represents the ratio of proteins with scTM > 0.5, reflecting the designability. '$\text{RMSD}^{1.0}_{\text{motif}}$' indicates the ratio of proteins with $\text{RMSD}_{\text{motif}} <$ 1Å. 'SR' represents the success rate metric. The diversity of the protein designs is quantified by 'Div.'. Finally, 'pdbTM$_{0.7}$' accounts for the ratio of proteins with $\text{pdbTM} < 0.7$ and scTM > 0.5 , assessing their novelty relative to existing structures in the PDB database. Details are described in Sec.A.5.

## 5.3 MAIN RESULTS

**Double motifs.** We initially evaluate MoDiff under two motif settings: $\mathcal{M}^{\text{unknown}}$ and $\mathcal{M}^{\text{exist}}$, as illustrated in Tab.1. Subsequently, we visualize the success rates for a subset of deign tasks in $\mathcal{M}^{\text{exist}}$,

Table 1: **Average metrics across all benchmarks.**

|  | $\text{scTM}_{0.5}$ ↑ | $\text{RMSD}_{\text{motif}}^{1.0}$ ↑ | SR ↑ | Div. ↑ | $\text{pdbTM}_{0.7}$ ↑ |
|---|---|---|---|---|---|
| $\mathcal{M}^{\text{exist}}$ | 65.92% | 31.50% | 19.21% | 93.11% | 29.07% |
| $\mathcal{M}^{\text{unknown}}$ | 60.83% | 29.81% | 16.35% | 96.83% | 31.99% |

presented in Fig.5. As depicted in Tab.1, MoDiff demonstrates favorable success rates in both settings. However, we observe a slightly higher success rate in $\mathcal{M}^{\text{exist}}$ compared to $\mathcal{M}^{\text{unknown}}$, indicating that certain motif combinations might pose challenges in coexisting within the same protein.

**In-depth analysis.** We conduct a statistical analysis of the success rates in relation to motif features, encompassing solvent-accessible surface area (Connolly, 1983), the average number of hydrogen bonds per residue, and the average number of contacts per residue. To account for the influence of motif features on design success rates, we fit the data using cubic polynomial curves with regularization. In Fig. 4, we observe that a higher number of contacts and hydrogen bonds per residue inside

Table 2: **Protein novelty enhancement driven by motif conditions.** VFN-Diff serves as the unconditioned baseline, while MoDiff introduces multi-motif conditions on top of VFN-Diff. $\mathcal{M}^{\text{exist}}$ and $\mathcal{M}^{\text{unknown}}$ represent whether the given motifs originate from the same protein or different proteins. Only successfully designed proteins are included in the statistics.

|  | VFN-Diff | MoDiff $\mathcal{M}^{\text{exist}}$ | MoDiff $\mathcal{M}^{\text{unknown}}$ |
|---|---|---|---|
| $\text{pdbTM}_{0.7}$ ↑ | 24.90% | 29.07% | **31.99%** |

the motif correlated with higher design success rates. Conversely, larger motif surface areas are associated with lower success rates, indicating increased difficulty in design. Our results support the quantitative observations by Didi et al. (2023) regarding single-motif scaffolding. They find that alpha-helix motifs, notable for their compact structure with reduced surface area coupled with a denser network of hydrogen bonds and contacts, are scaffolded more easily when compared to motifs of other SCOPe (Fox et al., 2014) classes. Further analysis can be found in Sec. A.6.

**Novelty related to the given motifs.** In Tab. 2, we assess the novelty of the protein structures designed by MoDiff in comparison to the unconditional baseline model, VFN-Diff, utilizing the ratio of proteins with pdbTM scores below 0.7 as a metric. MoDiff and VFN-Diff are aligned in their experiment settings, sharing the same noise schedule and sample length. Within this controlled experimental setup, MoDiff demonstrates a distinct advantage in generating novel structures, as indicated by its higher $\text{pdbTM} < 0.7$ ratio, particularly with test samples in $\mathcal{M}^{\text{unknown}}$. The motifs in $\mathcal{M}^{\text{unknown}}$ originate from two distinct proteins. The enhanced novelty is due to the potential absence of proteins with the targeted motifs ($\mathcal{M}^{\text{unknown}}$) in the Protein Data Bank (PDB), increasing the chances that MoDiff's designs are unique among cataloged protein structures.

**More motifs.** MoDiff can be directly extended to accommodate more motifs and handle more complex scenarios. Here, we present cases with three to four motifs, as illustrated in Fig. 3. To the best of our knowledge, this is the first method capable of addressing the motif scaffolding problem with such a multitude of motifs.

## 5.4 ABLATION STUDY

In this subsection, we first analyze the performance improvements brought by our proposed enhancement modules, IMM and $\mathcal{L}_{\text{MR}}$, and then compare the impact of different diffusion hyperparameters. In Tab.3, we conduct ablation experiments on $\mathcal{M}^{\text{exist}}$. The experiments demonstrate that the two enhancement modules individually contribute to approximately 10% and 14% increases in success rate. The reason for this improvement lies in the fact that these mod-

Table 3: **Ablation study on the Motif Reconstruction loss ($\mathcal{L}_{\text{MR}}$) and Implicit Matching Module (IMM).**

| $\mathcal{L}_{\text{MR}}$ | IMM | $\text{scTM}_{0.5}$ ↑ | $\text{RMSD}_{\text{motif}}^{1.0}$ ↑ | SR ↑ |
|---|---|---|---|---|
| ✗ | ✗ | **93.21%** | 1.35% | 1.22% |
| ✗ | ✓ | 91.35% | 6.67% | 5.77% |
| ✓ | ✗ | 91.28% | 2.24% | 2.18% |
| ✓ | ✓ | 60.83% | **29.81%** | **16.35%** |

ules enable MoDiff to better replicate the given motifs, leading to an increased proportion of

$\mathrm{RMSD}_{\mathrm{motif}} < 1\text{Å}$. However, on the other hand, although motifs can appear more stably in the designed proteins, the designability of the generated proteins slightly decreases. Tab.4 analyzes the performance of MoDiff under different diffusion hyperparameter settings. Specifically, in settings with lower noise, designability is enhanced, but $\mathrm{RMSD}_{\mathrm{motif}} < 1\text{Å}$ performance is relatively poorer.

## 6 CONCLUSIONS

The multi-motif scaffolding problem is a critical issue in protein design, contributing to the development of vaccines and enzymes. Unlike RFdiffusion, which requires expert knowledge to specify motif positions, MoDiff achieves the automatic assembly of multiple motifs into the same protein. In silico experiments demonstrate that MoDiff can achieve a high success rate in designing diverse motifs, even in the absence of expert knowledge. The generated scaffolds by MoDiff ex-

Table 4: **Ablation study on diffusion noise scale and the numbers of diffusion steps.**

| Noise Scale | 1.0 | 0.5 | 0.1 | 0.1 |
|---|---|---|---|---|
| Num. Step | 500 | 500 | 500 | 100 |
| mean scTM ↑ | 0.51 | 0.58 | **0.59** | 0.57 |
| $\mathrm{scTM}_{0.5}$ ↑ | 39.36 % | 59.87% | **60.83%** | 55.13% |
| $\mathrm{RMSD}_{\mathrm{motif}}^{1.0}$ ↑ | **38.72%** | 36.73% | 29.81% | 33.08% |
| SR ↑ | 16.15% | **23.65%** | 16.35% | 19.42% |

hibit significant diversity. Furthermore, by specifying motifs ($\mathcal{M}^{\mathrm{unknown}}$) from different proteins to MoDiff, the novelty of the proteins generated in this scenario is notably superior to the unconditional baseline, VFN-Diff. This is because, in such cases, proteins with given motifs ($\mathcal{M}^{\mathrm{unknown}}$) may not be present in the Protein Data Bank (PDB), increasing the probability that the designed protein is distinct from all known proteins. However, while MoDiff may become a general solution to the multi-motif scaffolding problem, the challenge of ensuring that multiple motifs exert the desired biological functions remains unresolved and requires further exploration.

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

# A    APPENDIX

## A.1    LIMITATIONS

The problem of assembling motifs to achieve specific functionalities remains unresolved. In proteins such as enzymes, motifs need to be assembled in a specific manner to realize particular biological functions. In this study, while MoDiff successfully automates the assembly of different motifs, the functionality resulting from the assembly of these motifs has not been explored. This aspect involves wet-lab experimental testing and is left for future work.

Generating long proteins is challenging. Incorporating more motifs results in longer protein lengths. However, both VFN-Diff and RFdiffusion exhibit noticeable limitations in generating proteins exceeding 500 amino acids, with a significant decrease in designability. This limitation hinders the success rate of MoDiff in assembling more motifs, such as five motifs, leading to lower success rates. Addressing the unresolved issue of enhancing the diffusion-based model to generate long proteins remains a critical area for future research. Further exploration is needed in future work to tackle this challenge.

## A.2    IMPLEMENT DETAILS

The model described in our study features a backbone architecture equivalent to the VFN-Diff model for consistent performance, coupled with a motif module designed to enrich the representation of protein features. The motif module comprises a VFN module with 4 blocks, a self-attention mechanism with 2 blocks and 8 heads, and a cross-attention component with 2 blocks to integrate motif and backbone information. Additionally, the motif node embedding size is set to 128.

## A.3    TRAINING DETAILS

**Hyper parameters.** In our training protocol, we employ the Adam optimizer with a learning rate set to 0.0001, and momentum hyperparameters $\beta_1$ and $\beta_2$ at 0.9 and 0.999, respectively. We maintain a batch size of 64. For the loss function, we weight the motif fape loss $\mathcal{L}_{\text{MR}}$ at 3 and the matching loss $\mathcal{L}_{\text{matching}}$ at 10. All other settings are consistent with those outlined in the VFN-Diff.

## A.4    MOTIF DATASET DETAILS

The motif dataset for this study originates from PROSITE, curated by the Swiss Institute of Bioinformatics (SIB), with 16,251 motif fragments aligned to PDB motifs. PROSITE, updated as of January 24, 2024, comprises 1942 documentation entries, 1311 patterns, and 1400 ProRules, encompassing various protein sequence recognition tools.

To evaluate MoDiff, motifs from protein which is already present in the training are excluded. For insights into MoDiff's performance on more complex scaffolds, additional motifs used for triple or quad configurations are documented in Tab. 7. Tab. 6 provides details for $\mathcal{M}^{\text{unknown}}$ and Tab. 5 for $\mathcal{M}^{\text{exist}}$. Within the tables, the term "Motif" corresponds to the Accession ID from PROSITE, "PDB" denotes the protein in which the motif was identified, "Input" refers to the specific chain and the start and stop residues of the motif on the protein, and "Length" denotes the number of amino acid residues constituting the motif. For $\mathcal{M}^{\text{exist}}$, different motifs under the same PDB ID are grouped together as one set of inputs; for $\mathcal{M}^{\text{unknown}}$, different motifs associated with the same Task ID are compiled as a single input set.

Table 5: Double Motif Scaffolding Benchmark $\mathcal{M}^{\text{exist}}$

| Motif | PDB | Input | Length | Description |
|---|---|---|---|---|
| PS00098 | 1PXT | A121-139 | 19 | Thiolases acyl-enzyme intermediate signature. |
| PS00099 | 1PXT | A398-411 | 14 | Thiolases active site. |
| PS00137 | 6RUH | A69-79 | 11 | Serine proteases, subtilase family, histidine active site. |
| PS00138 | 6RUH | A222-232 | 11 | Serine proteases, subtilase family, serine active site. |
| PS00181 | 1HTO | A258-273 | 16 | Glutamine synthetase putative ATP-binding region signature. |
| PS00182 | 1HTO | A385-397 | 13 | Glutamine synthetase class-I adenylation site. |
| PS00220 | 8GVE | A829-843 | 15 | Anion exchangers family signature 2. |
| PS00219 | 8GVE | A682-693 | 12 | Anion exchangers family signature 1. |
| PS00742 | 5JVN | A764-782 | 19 | PEP-utilizing enzymes signature 2. |
| PS00370 | 5JVN | A451-462 | 12 | PEP-utilizing enzymes phosphorylation site signature. |
| PS00407 | 6UVS | A53-66 | 14 | Connexins signature 1. |
| PS00408 | 6UVS | A169-185 | 17 | Connexins signature 2. |
| PS00900 | 7ZC4 | A918-929 | 12 | Bacteriophage-type RNA polymerase family active site signature 1. |
| PS00489 | 7ZC4 | A985-999 | 15 | Bacteriophage-type RNA polymerase family active site signature 2. |
| PS00490 | 1R27 | A776-793 | 18 | Prokaryotic molybdopterin oxidoreductases signature 2. |
| PS00551 | 1R27 | A47-65 | 19 | Prokaryotic molybdopterin oxidoreductases signature 1. |
| PS00630 | 6B63 | A199-213 | 15 | Inositol monophosphatase family signature 2. |
| PS00629 | 6B63 | A79-92 | 14 | Inositol monophosphatase family signature 1. |
| PS00675 | 5EXP | A170-183 | 14 | Sigma-54 interaction domain ATP-binding region A signature. |
| PS00676 | 5EXP | A232-247 | 16 | Sigma-54 interaction domain ATP-binding region B signature. |
| PS60032 | 1JS4 | A49-66 | 18 | Glycosyl hydrolases family 9 (GH9) active site signature 1. |
| PS00698 | 1JS4 | A413-431 | 19 | Glycosyl hydrolases family 9 (GH9) active site signature 3. |
| PS00816 | 3MI3 | A221-234 | 14 | Alpha-isopropylmalate and homocitrate synthases signature 2. |
| PS00815 | 3MI3 | A42-58 | 17 | Alpha-isopropylmalate and homocitrate synthases signature 1. |
| PS00882 | 4GD3 | A19-36 | 18 | Nickel-dependent hydrogenases b-type cytochrome subunit signature 1. |
| PS00883 | 4GD3 | A181-198 | 18 | Nickel-dependent hydrogenases b-type cytochrome subunit signature 2. |
| PS00978 | 2QCU | A352-362 | 11 | FAD-dependent glycerol-3-phosphate dehydrogenase signature 2. |
| PS00977 | 2QCU | A9-26 | 18 | FAD-dependent glycerol-3-phosphate dehydrogenase signature 1. |
| PS01061 | 6R6B | A188-200 | 13 | Flagella transport protein fliP family signature 2. |
| PS01060 | 6R6B | A140-155 | 16 | Flagella transport protein fliP family signature 1. |
| PS01090 | 1XWY | A123-133 | 11 | TatD deoxyribonuclease family signature 2. |
| PS01091 | 1XWY | A189-205 | 17 | TatD deoxyribonuclease family signature 3. |
| PS01126 | 3AVT | A12-27 | 16 | Elongation factor Ts signature 1. |
| PS01127 | 3AVT | A75-85 | 11 | Elongation factor Ts signature 2. |
| PS01174 | 2ZSH | A185-197 | 13 | Lipolytic enzymes "G-D-X-G" family, putative serine active site. |
| PS01173 | 2ZSH | A109-125 | 17 | Lipolytic enzymes "G-D-X-G" family, putative histidine active site. |
| PS01183 | 4OBW | A78-93 | 16 | ubiE/COQ5 methyltransferase family signature 1. |
| PS01184 | 4OBW | A215-229 | 15 | ubiE/COQ5 methyltransferase family signature 2. |
| PS01348 | 4J72 | A187-198 | 12 | MraY family signature 2. |
| PS01347 | 4J72 | A70-82 | 13 | MraY family signature 1. |

Table 6: Double Motif Scaffolding Benchmark $\mathcal{M}^{\text{unknown}}$

| Task | Motif | PDB | Input | Length | Description |
|---|---|---|---|---|---|
| 0 | PS00220 | 8GV9 | A829-843 | 15 | Anion exchangers family signature 2. |
| 0 | PS00769 | 7YBR | A105-117 | 13 | Transthyretin signature 2. |
| 1 | PS50837 | 7VTP | A518-536 | 19 | NACHT-NTPase domain profile. |
| 1 | PS00566 | 7SE9 | A185-199 | 15 | Fibrillarin signature. |
| 2 | PS00219 | 8GVH | A682-693 | 12 | Anion exchangers family signature 1. |
| 2 | PS00295 | 8GP3 | A61-79 | 19 | Arrestins signature. |
| 3 | PS00606 | 6U5W | A1294-1310 | 17 | Ketosynthase family 3 (KS3) active site signature. |
| 3 | PS01009 | 3MZ8 | A114-124 | 11 | CRISP family signature 1. |
| 4 | PS00885 | 7TBU | A742-760 | 19 | EPSP synthase signature 2. |
| 4 | PS00650 | 7SF7 | A1093-1108 | 16 | G-protein coupled receptors family 2 signature 2. |
| 5 | PS00140 | 7ZM0 | A84-100 | 17 | Ubiquitin carboxyl-terminal hydrolase family 1 cysteine active-site. |
| 5 | PS00376 | 7RW5 | A131-141 | 11 | S-adenosylmethionine synthase signature 1. |

Table 7: Additional Motif Details

| Motif | PDB | Input | Length | Description |
|---|---|---|---|---|
| PS00739 | 7ZD8 | A214-230 | 17 | S-adenosyl-L-homocysteine hydrolase signature 2. |
| PS00250 | 7Y1R | A282-297 | 16 | TGF-beta family signature. |
| PS00962 | 7WU0 | A15-26 | 12 | Ribosomal protein S2 signature 1. |
| PS00665 | 7KM0 | A41-58 | 18 | Dihydrodipicolinate synthase signature 1. |
| PS00613 | 7KBU | A523-533 | 11 | Osteonectin domain signature 2. |
| PS00749 | 7DSA | A256-266 | 11 | F-actin capping protein alpha subunit signature 2. |
| PS00248 | 6YW8 | A67-80 | 14 | Nerve growth factor family signature. |
| PS00095 | 6W8V | A1576-1594 | 19 | C-5 cytosine-specific DNA methylases C-terminal signature. |
| PS00062 | 6TUF | A144-161 | 18 | Aldo/keto reductase family signature 2. |
| PS00381 | 6N80 | A90-101 | 12 | Endopeptidase Clp serine active site. |
| PS00751 | 6KRD | A62-78 | 17 | Chaperonins TCP-1 signature 2. |
| PS00973 | 6IIK | A418-436 | 19 | Ubiquitin specific protease (USP) domain signature 2. |
| PS60003 | 8IO8 | A162-180 | 19 | Phosphoketolase signature 2. |
| PS00219 | 8GVH | A682-693 | 12 | Anion exchangers family signature 1. |
| PS00220 | 8GV9 | A829-843 | 15 | Anion exchangers family signature 2. |
| PS00310 | 8ATH | A40-54 | 15 | Lysosome-associated membrane glycoproteins duplicated domain signature. |
| PS00650 | 7SF7 | A1093-1108 | 16 | G-protein coupled receptors family 2 signature 2. |
| PS00566 | 7SE9 | A185-199 | 15 | Fibrillarin signature. |
| PS00785 | 7PBY | A29-41 | 13 | 5'-nucleotidase signature 1. |
| PS00796 | 7O57 | A41-51 | 11 | 14-3-3 proteins signature 1. |
| PS00750 | 7X7Y | A33-45 | 13 | Chaperonins TCP-1 signature 1. |
| PS00237 | 7WF7 | A130-146 | 17 | G-protein coupled receptors family 1 signature. |
| PS01255 | 7UAI | A87-96 | 10 | Fetuin family signature 2. |
| PS00726 | 7TR7 | A89-98 | 10 | AP endonucleases family 1 signature 1. |
| PS00263 | 8S9Y | A9-25 | 17 | Natriuretic peptides signature. |
| PS00509 | 8E20 | A1386-1400 | 15 | Ras GTPase-activating proteins domain signature. |
| PS01307 | 8BRI | A182-199 | 18 | Flagellar motor protein motA family signature. |
| PS00930 | 7ZW1 | A207-222 | 16 | Peripherin / rom-1 signature. |

## A.5 METRIC DETAILS

**Motif RMSD.** As defined in Eq. 1, $\text{RMSD}_{\text{motif}}$ assesses whether the given motif exists in the protein structure.

**scTM.** scTM represents the protein's designability, *i.e.* whether the generated protein structure is feasible. Specifically, the generated structures are fed into ProteinMPNN (Dauparas et al., 2022) to predict the corresponding sequences, which are then utilized by ESMFold (Hie et al., 2022) for structure prediction. If the structure predicted by ESMFold is similar to the generated structure (high scTM), it implies the generated structure is feasible.

**Success rate.** Success rate signifies the proportion of successful designs, where motifs are present ($\text{RMSD}_{\text{motif}} < 1\text{Å}$), and the protein is designable ($\text{scTM} > 0.5$).

**Diversity:** For assessing the diversity of the generated protein sequences, we follow the methodology outlined in (Yim et al., 2023) and utilized MaxCluster (Herbert & Sternberg, 2008) for hierarchical clustering of protein backbones. A selection criterion, $\text{scTM} > 0.5$, was impose for cluster consideration to minimize the influence of proteins with low designability on the diversity metric. We select a single protein with the highest scTM score from each diffusion sample, generated by ESMFold, for inclusion. The diversity metric is then calculated as the ratio of the number of clusters to the total diffusion samples, $N_{diff}$, ensuring a more stringent and relevant evaluation.

**pdbTM:** Novelty in protein design is quantified by comparing the highest-scoring proteins from ESMFold-generated PDB files against the PDB database using Foldseek (van Kempen et al., 2023). Proteins are required to have an scTM score greater than 0.5 to be included in this analysis. The pdbTM score is used to measure structural novelty, with proteins considered novel at a threshold of $\text{pdbTM} < 0.7$, referred to as $\text{pdbTM}_{0.7}$. This cutoff, less stringent than the $\text{pdbTM} < 0.6$ criterion used in (Yim et al., 2023), allows for the recognition of a wider range of novel protein structures.

A.6   ADDTIONAL QUALITATIVE ANALYSIS

**Analysis of motif features' influence.** In Sec. 5.3, we investigate the relationship between the success rate and various motif features: the number of hydrogen bonds per residue, the number of contacts per residue, and the solvent-accessible surface area (SASA). Our findings reveal a positive correlation between both the number of hydrogen bonds per residue and the number of contacts per residue with the success rate, while the solvent-accessible surface area exhibits a negative correlation with the success rate.

Building on this analysis, we further explore the impact of these motif features on scTM and $\mathrm{RMSD_{motif}} > 1$ ratio in Fig. 4. Interestingly, we observe no significant correlation between the motif features and scTM. However, there is a positive correlation between both the number of hydrogen bonds per residue and the number of contacts per residue with $\mathrm{RMSD_{motif}} > 1$ ratio, and a negative correlation between solvent-accessible surface area and $\mathrm{RMSD_{motif}} > 1$ ratio.

These results suggest that the difficulty of a motif, as reflected by $\mathrm{RMSD_{motif}} > 1$ ratio, predominantly influences the success rate. This also implies that the bottleneck of the MoDiff model lies in encoding complex motifs and generating them with high precision under specified conditions.

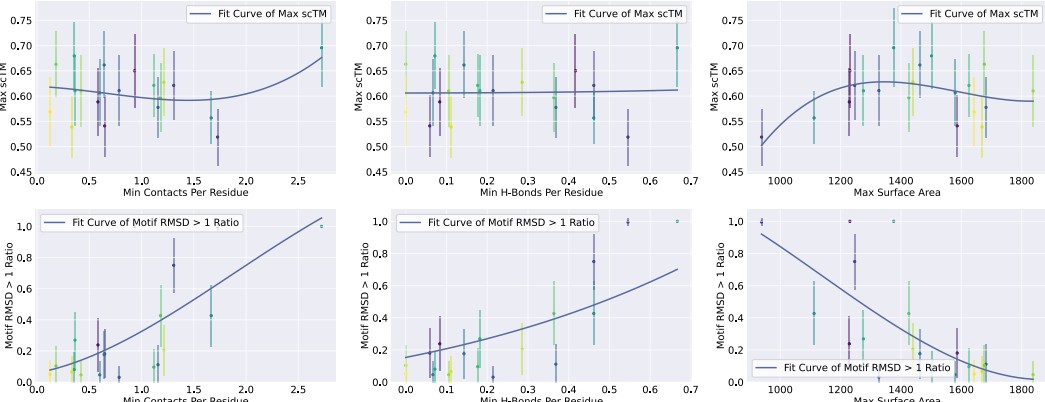

Figure 6: **Deep analysis of 'scTM$_{0.5}$' and 'RMSD$_{\mathrm{motif}}^{1.0}$' *w.r.t* motif features.** 'scTM$_{0.5}$' represents the ratio of proteins with scTM scores greater than 0.5, reflecting the designability. 'RMSD$_{\mathrm{motif}}^{1.0}$' metric indicates the ratio of proteins with $\mathrm{RMSD_{motif}}$ less than 1Å. We conduct separate analyses on hydrogen bonds number per residue, contacts number per residue, and solvent-accessible surface area to assess their impact on the design success rate and various influencing factors.

**Qualitative analysis of scTM in different ranges.** In this section, we follow SMCDiff (Trippe et al., 2022) to offer insights into backbone designs and ESMFold predictions across different scTM values, facilitating the interpretation of scTM metrics we previously discussed. Fig. 7 explores a categorization of scTM into four ranges:

- The first row corresponds to backbone designs with scTM $\geq 0.8$. The backbone designs are observed to closely align with the ESMFold predictions.
- The second row corresponds to designs with $0.6 \leq \mathrm{scTM} < 0.8$. In this tier, ESMFold predictions and backbone designs have a high degree of similarity, but there are subtle differences and inconsistencies in flexible regions.
- The third row presents designs with $0.4 \leq \mathrm{scTM} < 0.6$. These designs represent the boundary of what we consider designable (scTM $> 0.5$). Here, the ESMFold predictions maintain the general shape of the backbone design, though differences may exist in the ordering and composition of secondary structures.
- The final category is for scTM $< 0.4$, indicative of failure cases where the scTM is low. In these instances, ESMFold predictions include numerous disordered regions and show little structural resemblance to the original backbone design.

**Qualitative analysis of $\mathrm{RMSD_{motif}}$ in different ranges.** In this section, we analyze the $\mathrm{RMSD_{motif}}$ metric previously discussed. Fig. 8 categorizes $\mathrm{RMSD_{motif}}$ into three ranges:

Figure 7: Qualitative analysisof scTM in different ranges. The light blue visualizations display backbone designs generated by MoDiff, along with their respective sequence lengths. In contrast, the golden visualizations highlight the top-scoring ESMFold (Hie et al., 2022) predictions based on the scTM metric for sequences generated by ProteinMPNN (Dauparas et al., 2022) that correspond to the backbone designs visualized in light blue. The first column illustrates the alignment results between the second and third columns and provides the specific PDB ID for each design tasks.

- The first row represents cases where the root-mean-square deviation (RMSD) between the conditionally provided motif and the model-designed backbone is less than 1. Under these circumstances, the motif and backbone are observed to align very well, which is the standard we consider indicative of a successful motif-scaffold pairing.
- The second row illustrates instances where the $\text{RMSD}_{\text{motif}}$ metric is between 1 and 2, where the motif generally aligns with the backbone, but differences and discrepancies are noticeable in the details of flexible regions, leading us to categorize these as design failures.
- The third row showcases cases where the $\text{RMSD}_{\text{motif}}$ metric exceeds 2, highlighting a significant divergence between the motif and the backbone. For instance, within the 5EXP design task, the alpha-helix region exhibits alignment difficulties, indicating a substantial discrepancy in the structural fidelity of this motif when compared to the intended design.

## A.7 ADDTIONAL MOTIF-SCAFFOLDING RESULTS

**Known relative poses between motifs.** Previous methods, such as RFDiffusion (Watson et al., 2023), are capable of handling scenarios where the relative poses between motifs are known. Although MoDiff implicitly assigns motifs to the protein backbone, thereby automating the design of relative motif positions, our approach can still accommodate situations where the relative poses between motifs are known. This is achieved by employing the same VFN module to encode a set of motifs, as demonstrated by the cases outlined in Tab. 8.

Table 8: Results with given relative poses between motifs. 'scTM$_{0.5}$' represents the ratio of proteins with scTM scores greater than 0.5, reflecting the designability. '$\text{RMSD}_{\text{motif}}^{1.0}$' metric indicates the ratio of proteins with $\text{RMSD}_{\text{motif}}$ less than 1Å. 'SR' represents the success rate.

| PDB | scTM$_{0.5}$ ↑ | $\text{RMSD}_{\text{motif}}^{1.0}$ ↑ | SR ↑ |
|---|---|---|---|
| 7F68 | 91.92% | 47.69% | 42.31% |
| 2OE4 | 68.85% | 27.31% | 16.92% |
| 7Q7A | 75.77% | 16.53% | 11.54% |

**Large scaffold size.** Constrained by the limitations of the unconditional generative model, there is a decline in the self-consistency metric as the length of the generated protein increases, particularly

| RMSD | 5EXP, PS00676 | | 5JVN, PS00742 | |
|---|---|---|---|---|
| < 1.0 | motifRMSD=0.58 | motifRMSD=0.66 | motifRMSD=0.56 | motifRMSD=0.64 |
| [1.0, 2.0] | motifRMSD=1.30 | motifRMSD=1.71 | motifRMSD=1.12 | motifRMSD=1.46 |
| > 2.0 | motifRMSD=2.14 | motifRMSD=2.27 | motifRMSD=2.13 | motifRMSD=2.45 |

Figure 8: Qualitative analysis of $\mathrm{RMSD}_{\mathrm{motif}}$ in different ranges. The visualization includes samples from two motif-scaffold tasks, with the corresponding PDB ID and Motif ID associated with $\mathcal{M}^{\mathrm{exist}}$, demonstrating alignments across a spectrum of $\mathrm{RMSD}_{\mathrm{motif}}$ values.

when the protein backbone exceeds 400 residues in length. Nevertheless, our model retains the capability to design proteins with larger scaffold sizes, as demonstrated by the following cases in Fig. 9.

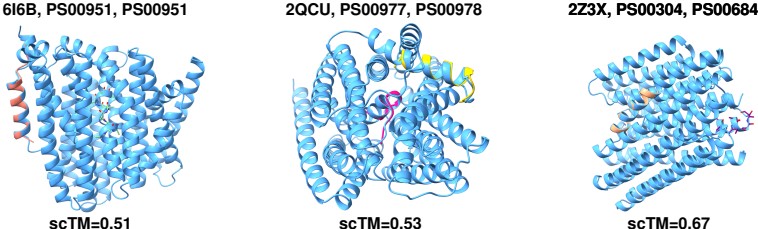

Figure 9: Visualization of alignment results between motifs and backbones for three successful cases of large scaffold sizes, each 500 amino acids in length. Each subplot highlights the precise positioning of motifs ($\mathrm{RMSD}_{\mathrm{motif}} < 1\text{Å}$), labels the name of the design task, and indicates the scTM score, signifying that the proteins are designable (scTM > 0.5).

**Additional success cases.** Similar to the enumeration of triple and quadruple motif scaffold cases in Sec. 5.3, Fig. 10 extends the catalogue to include additional success cases involving double motif scaffolds.

**Failure cases.** As discussed previously, MoDiff exhibits improved performance with motifs characterized by a higher proportion of secondary structures. In contrast, when confronted with motifs where flexible regions such as loops predominate, there is a decrease in scTM and motif rmsd. The cases in Fig. 11 highlighted below indicate that the generation process may encounter challenges in achieving the desired level of detail when dealing with intricate motifs. In subsequent work, we aim to enhance the matching module and increase the model's capacity to overcome this limitation, thereby further advancing motif-scaffolding performance.

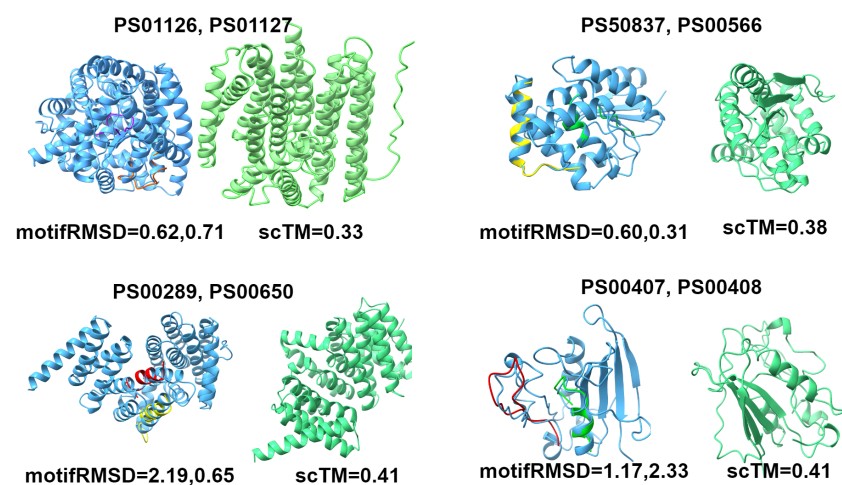

Figure 10: Visualization of failure cases. Each subplot illustrates the deviation of motifs with a high root-mean-square deviation ($\text{RMSD}_{\text{motif}} > 1\text{Å}$) and marks the design task where the structural alignment did not meet the criteria for designability, as reflected by low scTM scores ($\text{scTM} \leq 0.5$).

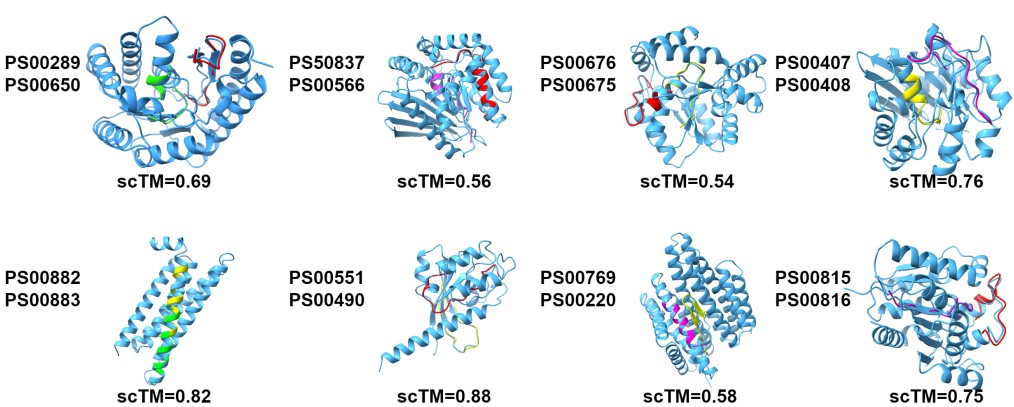

Figure 11: Further examples of successful motif integration are presented, with each subplot emphasizing the precise alignment of motifs ($\text{RMSD}_{\text{motif}} < 1\text{Å}$). The subplots also annotate the respective design tasks and display the scTM score, denoting the proteins' designability with high confidence ($\text{scTM} > 0.5$).

