# OpenReview forum: "MoDiff: a Protein Diffusion Model for Multi-motif Scaffolding Problem"
_ICLR.cc/2025/Conference — Submitted to ICLR 2025_

### Official Review · Reviewer_yHes · 2024-10-31

**Soundness:** 2
**Presentation:** 2
**Contribution:** 2
**Rating:** 3
**Confidence:** 5

**Summary:**

This paper presents a novel protein diffusion model named MoDiff, aimed at solving the multi-motif scaffold design problem. Traditional methods are generally based on the inpainting paradigm, requiring the relative positions of motifs to be known, which limits their applicability. MoDiff automatically determines the positions of motifs within the protein backbone through the diffusion process, eliminating the need for manual specification of relative positions, especially when motif positions are unknown. Experimental results indicate that MoDiff can generate scaffolds containing multiple specified motifs and shows reasonable success in protein design tasks. The study also introduces an implicit matching module and motif reconstruction loss to improve model performance. MoDiff offers a new approach to multi-motif scaffold design.

**Strengths:**

This paper provides a new modeling approach and perspective for multi-motif scaffolding in protein design. In the original SE(3) diffusion framework, a cross-attention module that incorporates motif embeddings is added to effectively integrate multi-motif information. The feasibility of this method is verified through experimental results on a benchmark dataset.

**Weaknesses:**

1. The paper lacks key details about the algorithm, and the descriptions are not clear, leading to confusion. For instance, it does not mention how multi-motifs, extracted from a system with known relative orientations and positions, maintain these relative features during decoding. Moreover, the definition of the label $\textbf{C}$ in the cross-attention module is not clearly explained.
2. There are few comparative methods and experimental results, and the paper lacks evaluations using the most basic motif-scaffolding benchmark provided by RFdiffusion.
3. The presentation is mediocre. For example, there is insufficient description of the VFN network framework, and essential features such as residue index and motif mask are not adequately explained.

**Questions:**

1. In the IMM module, does N represent the total length of the multi-motif and scaffolding, or just the scaffolding length? How is the label $\textbf{C}$ used for supervising the cross-attention defined? What is the physical meaning of this supervision on the 2D distance map?
2. If the relative orientations and positions between motifs are known, how is this information utilized and how is their relative positioning preserved during generation?
3. For single-motif scaffolding tasks, how does the performance compare to RFdiffusion on the benchmark?
4. Generally, the function of a motif is highly dependent on both its structure and its side-chain sequence. Therefore, most studies retain the motif sequence. Why is this aspect not considered here?
5. How can it be verified that injecting motifs in the latent space is superior in terms of direct physical meaning compared to using a 2D distogram as a template (e.g., RFdiffusion for VHH, GENIE2)?

---

### Official Review · Reviewer_LbEW · 2024-11-01

**Soundness:** 2
**Presentation:** 3
**Contribution:** 3
**Rating:** 5
**Confidence:** 4

**Summary:**

This paper addresses the multi-motif scaffolding problem. There are two main contributions: the implicit matching module, and the motif reconstruction loss.

**Strengths:**

This paper explains clearly the importance of the multi-motif scaffolding problem for protein engineering and the limitations of existing methods. The two introduced improvements appear to improve performance on this task. The implicit is an neat, intuitive, and novel extension of cross attention that addresses the particular challenges of scaffolding multiple motifs. The breakdown of success rate for the design tasks on $M^\text{exist}$ is informative. The paper is well presented.

**Weaknesses:**

One of the central claims of the paper is that cross attention is insufficient and the implicit matching module (IMM) is a necessary development. It is not clear from Section 5.4 whether the ablation of IMM replaces it with cross attention, or removes the conditioning entirely - a much easier baseline to beat. For this reason I am recommending rejection, but the paper could be improved by demonstrating the insufficiency of cross attention more clearly.

The paper uses VFN-Diff, but it would be interesting to see if the ideas improve performance on other SE(3) diffusion models. Although qualitative results are given for 3+ motifs, the paper could be improved by benchmarking the model on tasks with more motifs. The more in-depth analyses within this paper could also be repeated for the ablated methods.

Minor comment: line 239 refers to a section that does not exist.

**Questions:**

1. What is the difference in performance between cross attention and IMM?
2. Are these ideas presented here transferable to other protein diffusion models?
3. How does performance of MoDiff vary with the number of motifs provided?

---

### Official Review · Reviewer_Hayc · 2024-11-02

**Soundness:** 3
**Presentation:** 2
**Contribution:** 2
**Rating:** 3
**Confidence:** 3

**Summary:**

This paper proposes MoDiff, a protein diffusion model designed to address the multi-motif scaffolding problem, which is crucial in fields like drug discovery and enzyme design. Unlike existing methods, MoDiff can generate diverse and novelty scaffolds even when the relative positions between motifs are unknown (no expert knowledge) and enhances the success rate of scaffold design through an implicit matching module and motif reconstruction loss.

**Strengths:**

1. The authors introduce the multi-motif scaffolding problem and propose the MoDiff framework, which can generate diverse and novel scaffolds even when the relative positions between motifs are unknown.
2. MoDiff leverages prior knowledge learned from data distributions without expert knowledge and achieves high success rates through the implicit matching module and motif reconstruction loss components.

**Weaknesses:**

1. In the introduction, the explanation of existing methods (inpainting methods) for the multi-motif scaffolding problem is somewhat limited, and the inefficiency of traditional cross-attention methods is not fully explained. Additionally, some contributions could be better clarified.
-  Although the authors describe showcasing the promising prospects of deep learning approaches in the contribution, it feels somewhat awkward, as the introduction does not compare deep learning with traditional computational methods. It might be more natural to explain why deep learning is needed for this task or highlight the challenges of the multi-motif scaffolding problem and the importance of addressing it.

- The description of MoDiff’s model, core keywords, and advantages is somewhat unclear. A more precise explanation would help in highlighting the strengths of the method.

2. IMM is proposed to capture the motif correspondence (π_1,π_2), but based on the provided equations (10) and (11), it appears to compute and learn attention for each motif (M_1, M_2) separately in relation to the protein backbone representation. This approach doesn’t seem to directly address the relationship between M_1 and M_2. A more effective approach might involve jointly considering the attention for M_1 and M_2.

3. In section 5.3 "More motifs" part, MoDiff is described as the "first method," but FADiff [1] already addresses this problem. This claim may need rephrasing, and it would also be helpful to check if a comparative analysis with FADiff is possible.

4. The code has not yet been released, which, if provided, would enhance reproducibility and facilitate a better understanding of the method.

**Questions:**

1. Section 4 states that MoDiff "leverages prior knowledge learned from the data distribution". Could you provide an additional explanation on how this is achieved? A more detailed explanation in the introduction would also help clarify this key strength of the method.
2. In the methods section on p5, there is a symbol "??". It seems to be a latex mapping error, please check this.
3.  In section 4.2.2 ("The protein backbone branch"), it says that parameters are not shared across iterations. Does this mean a different model is used for each iteration? If so, could you explain why this approach was chosen?
4. (W3) In Table 2, it seems VFN-Miff was evaluated on M^exist. Are there other baselines available for comparison? Additionally, FADiff [1] is also reported to address the multi-motif scaffolding problem. Could you consider including performance comparisons with this model?
5. In section 5.4, it is mentioned that L_MR and IMM improved success rates by 10% and 14%, respectively. Could you clarify which results were used for this comparison? Does this mean that the experiments leveraging both L_MR and IMM achieved approximately 10% and 14% higher success rates than experiments where each component was tested individually?

[1] Liu et al. (2024) Floating Anchor Diffusion Model for Multi-motif Scaffolding, ICML

---

### Official Review · Reviewer_2x7s · 2024-11-03

**Soundness:** 3
**Presentation:** 3
**Contribution:** 3
**Rating:** 5
**Confidence:** 4

**Summary:**

This paper proposes MoDiff, a protein diffusion model built upon VFN-Diff specifically for multi-motif scaffolding, a novel task untackled by prior protein diffusion models. Novel implicit matching module is proposed, showing enhanced success in multi-motif scaffolding.

**Strengths:**

The paper is clear and easy to follow. The Implicit Matching Module is an interesting implementation to encourage the model to learn the matching between motifs and the protein. In addition, the multi-motif scaffolding task (with unknown relative spatial poses and sequential positions among motifs) is a novel construction and could have potential for the design of antibodies and enzymes.

**Weaknesses:**

One major concern with this paper is the lack of comparisons with other models. While MoDiff is designed to tackle a novel and more general task, it is crucial to compare MoDiff with other models on previously reported tasks (which are subtasks of the problem at hand) to provide a better understanding of model performance. For example,
- How well is MoDiff capturing the underlying protein structure space, which forms the basis for the success of multi-motif scaffolding?
- How does MoDiff perform in scaffolding a single motif (or the test set $M^{exist}$) compared to RFDiffusion? Though it might not be a completely fair comparison since the relative sequence and structure positions are unknown to MoDiff, it is helpful to understand if this additional flexibility helps with motif scaffolding (since the model could now optimize for motif positions).
- Recent work of Genie 2 (https://arxiv.org/abs/2405.15489) also attempts to tackle the problem of multi-motif scaffolding, though it maintains the relative position between motif sequences. It would be helpful to compare MoDiff to Genie 2 on the same set of multi-motif scaffolding tasks.

**Questions:**

- Motif sequence is crucial for the functionality of a motif. Is the motif sequence fixed in this design setting? Is the motif sequence information passed into the model? Is the motif sequence fixed in the ProteinMPNN sequence design?
- Figure 2A: It looks like a motif is defined as a continuous sequence of residues. Can a motif consist of multiple segments (with fixed relative positions between segments in the motif)? Does MoDiff support scaffolding with multiple motifs, where one or more of these motifs consist of multiple segments? This is important since residues involved in a functional motif could be close in the structural space but distant in the sequence (resulting in discontinuous segments in a motif).
- It would be helpful if the authors would provide a summary of VFN architecture in the Appendix and highlight the key differences between VFN and other frame-based architectures like IPA in AF2. What is the key motivation for the use of VFN and does it demonstrate better performance compared to other architectures? In addition, what is the architecture of SeqTrans? Is it a standard attention-based transformer?
- What is the rationale of using scTM > 0.5 as the designability metric instead of scRMSD < 2 that is more stringent and commonly used in recent motif scaffolding diffusion work like RFDiffusion?
- Section 5.2: What criteria are used for excluding PROSITE motifs present in the training dataset? How many motifs are there after this filtering process? Given that $M^{exist}$ and $M^{unknown}$ have a size of 20 and 6 respectively, what is the variation in both cases? How would the model perform for a different set of motif problems or for a larger set of motif problems?
- Section 5.3: When assessing novelty, does the PDB dataset contain PDB files from $M^{exist}$ and $M^{unknown}$?

---

### Meta-Review · Area_Chair_qaia · 2024-12-19

**Metareview:**

This paper proposes a novel protein diffusion model, called MoDiff, that takes the perspectives of multi-motif scaffolds to improve drug and enzyme design.
The reviewers note that the overall paper is written well, where the method and results are presented in a clear manner.
However, the reviewers also note that there is insufficient discussion and comparison against relevant existing methods and that the novelty and the efficacy of the proposed method haven't been sufficiently demonstrated convincingly.

**Additional Comments On Reviewer Discussion:**

The authors have not engaged with the reviewers during the discussion period and the reviewers concerns and doubts remain unaddressed.

---

### Decision · Program_Chairs · 2025-01-22

Reject